# Geographical drivers and climate-linked dynamics of Lassa fever in Nigeria

David W. Redding [1,2,8✉], Rory Gibb [2,8], Chioma C. Dan-Nwafor [3], Elsie A. Ilori[3], Rimamdeyati Usman Yashe[3], Saliu H. Oladele[3,4], Michael O. Amedu[3], Akanimo Iniobong[3], Lauren A. Attfield[1,2,5], Christl A. Donnelly [5,6], Ibrahim Abubakar [7], Kate E. Jones [1,2] & Chikwe Ihekweazu[3]

Lassa fever is a longstanding public health concern in West Africa. Recent molecular studies have confirmed the fundamental role of the rodent host (*Mastomys natalensis*) in driving human infections, but control and prevention efforts remain hampered by a limited baseline understanding of the disease's true incidence, geographical distribution and underlying drivers. Here, we show that Lassa fever occurrence and incidence is influenced by climate, poverty, agriculture and urbanisation factors. However, heterogeneous reporting processes and diagnostic laboratory access also appear to be important drivers of the patchy distribution of observed disease incidence. Using spatiotemporal predictive models we show that including climatic variability added retrospective predictive value over a baseline model (11% decrease in out-of-sample predictive error). However, predictions for 2020 show that a climate-driven model performs similarly overall to the baseline model. Overall, with ongoing improvements in surveillance there may be potential for forecasting Lassa fever incidence to inform health planning.

---

[1] Institute of Zoology, Zoological Society of London, Regent's Park, London NW1 4RY, United Kingdom. [2] Centre for Biodiversity and Environment Research, Department of Genetics, Evolution and Environment, University College London, Gower Street, London WC1E 6BT, United Kingdom. [3] Nigeria Centre for Disease Control, Abuja, Nigeria. [4] World Health Organisation, Abuja, Nigeria. [5] MRC Centre for Global Infectious Disease Analysis, School of Public Health, Imperial College London, W2 1PG London, United Kingdom. [6] Department of Statistics, University of Oxford, Oxford OX1 3LB, United Kingdom. [7] Institute of Global Health, University College London, Gower Street, London WC1E 6BT, United Kingdom. [8] These authors contributed equally: David W. Redding, Rory Gibb. ✉email: dwredding@gmail.com

Between 2018 and 2020, Nigeria recorded its highest annual incidences of Lassa fever (LF) to date (633 confirmed cases in 2018, 810 in 2019 and 1189 in 2020, across 29 states), prompting national and international healthcare mobilisation and raising concerns about an ongoing, systematic emergence of LF nationally[1,2]. Lassa virus (LASV; Arenaviridae, Order: Bunyavirales) is a WHO-listed priority pathogen and a major focus of international vaccine development funding[3] and, although often framed as a global health threat, LF is foremost a neglected endemic zoonosis. Lassa virus disease has a varied presentation, with many cases thought to be mild or asymptomatic[4]. More severe cases usually start with an unspecific fever and malaise, and occasionally progress to haemorrhagic symptoms, with fatalities in around 20% of severe cases. The disease has typically been characterised as having two main endemic foci in West Africa, one centred around Sierra Leone and Liberia, and the other in Nigeria, but in recent years most countries in the region have reported regular or sporadic cases[4]. Concurrently there has been a decline in surveillance-based case reports from the western hotspot around Sierra Leone, which may be related to the significant negative impacts of the recent Ebola epidemic on health systems and personnel. Conversely, Nigeria, which saw more limited impacts from Ebola, has continued to record a trend of increasing numbers of cases in this time.

The significant majority of observed LF cases—including those from recent years in Nigeria[5]—are thought to arise directly from spillover from the LASV reservoir host, the widespread synanthropic rodent *Mastomys natalensis*, although with hospital-acquired infections potentially occurring in small clusters of human-to-human transmission[6–8]. *M. natalensis* is the most abundant agricultural rodent pest in sub-Saharan Africa and is generally found in high numbers in many human-dominated land types, with lower abundance in natural and forested ecosystems. Populations of this species have a strong seasonal dynamic that varies across habitat types, likely related to changes in food availability over time[9]. Unintentional interactions with people occur in a variety of settings from cropland and households, as well as intentional contact through hunting-related practices. Risk factors for spillover, while not well understood, are thought to include factors that increase the direct and indirect

contact between rodents and people, including ineffective food storage, housing quality, and certain agricultural practices such as crop processing[10,11]. Evidence of correspondence between human case surges and seasonal rainfall patterns suggests that LF is a climate-sensitive disease[12], whose incidence may be increasing with regional climatic change[13].

The present-day incidence and burden, however, remain poorly defined, because LASV surveillance has historically been opportunistic or focused on known endemic districts with pre-existing diagnostic capacity[4], and often-cited annual case estimates (of up to 300,000) are consequently extrapolations based on limited serological evidence from a handful of early studies[14,15]. This, alongside LF's nonspecific presentation, means that many mild or subclinical infections (possibly 80% or more of infections) are thought to go undetected[16,17]. The patchy understanding of LF's true annual incidence and drivers hinders diagnosis, treatment and disease control[18] and provides a limited contextual understanding of whether the recent surges in reported cases have resulted from improvements in surveillance or a true emergence trend. To address these gaps, in this study we analyse the first long-term spatiotemporal epidemiological dataset of acute human LF case data, systematically collected over 8 years of surveillance in Nigeria. We use this dataset to characterise the epidemiology and spatial trends of LF in Nigeria between 2012 and 2019, and evaluate the drivers that explain the geographical distribution of LF occurrence and incidence. We then develop spatiotemporal predictive models to evaluate whether climatic variability can be used to predict interannual differences in the size and timing of outbreak peaks, to evaluate the scope for future forecasting of this high-burden disease.

## Results

**Recent trends in LF surveillance in Nigeria.** The dataset, collated by the Nigeria Centre for Disease Control (NCDC), consists of weekly epidemiological reports (WERs) of acute human LF cases collected by all 774 local government authorities (LGAs) across Nigeria between January 2012 and December 2019 (Fig. 1). Throughout the study period, 161 LGAs from 32 of 36 states reported cases, with a mean annual total of 276 (range 25 to 816) confirmed cases (for definitions see Supplementary Table 1), though with evidence of pronounced spatial and temporal clustering. For example, the majority of cases (~75%) are reported from just 3 of the 36 Nigerian states (Edo, Ondo and Ebonyi), with lower incidence overall in northern endemic states, in areas notably distant from diagnostic centres. There is consistent evidence of seasonality in all areas across the reporting period, except for 2014 to 2015, when a lull in recorded cases was coincident in timing with the West African Ebola epidemic (Supplementary Fig. 1). Annual dry season peaks of LF cases typically occur in January, confirming past hospital admissions data from Nigeria[19,20] and Sierra Leone[21], with some secondary peaks evident in early March and, increasingly, a small number of cases detected throughout the year (Fig. 1). Both overall temporal trends and cumulative case curves suggest that 2018 and 2019 appear to be markedly different from previous years, with very high peaks in confirmed cases extending from January into March, and high suspected case reporting continuing throughout 2019 (Fig. 1 and Supplementary Fig. 1).

Improvements to country-wide surveillance could, however, be driving any apparent increase in both the incidence and geographical extent of LF in Nigeria. For instance, during the 2012–2019 monitoring period, within-country LF surveillance and response was strengthened under NCDC coordination, with a dedicated NCDC Technical Working Group (LF TWG) established in 2016, the opening of three additional LF diagnostic

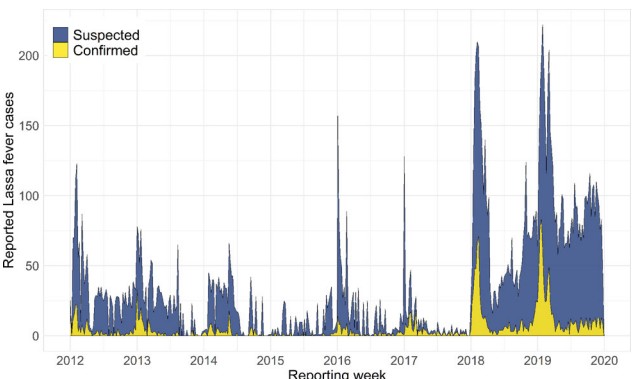

**Fig. 1 Temporal trends in country-wide Lassa fever case reporting from 2012 to 2019.** Polygon height shows the weekly total cases reported across Nigeria, with colour denoting the proportion of cases that were laboratory-confirmed (yellow) or suspected (blue). The full case time series was compiled from two reporting regimes at the Nigeria Centre for Disease Control: Weekly Epidemiological Reports 2012 to 2016, and Lassa Fever Technical Working Group Situation Reports 2017 to 2019 (with case reports followed-up to ensure more accurate counts; datasets shown separately in Supplementary Fig. 1). A full description of case definitions and reporting protocols is provided in Methods.

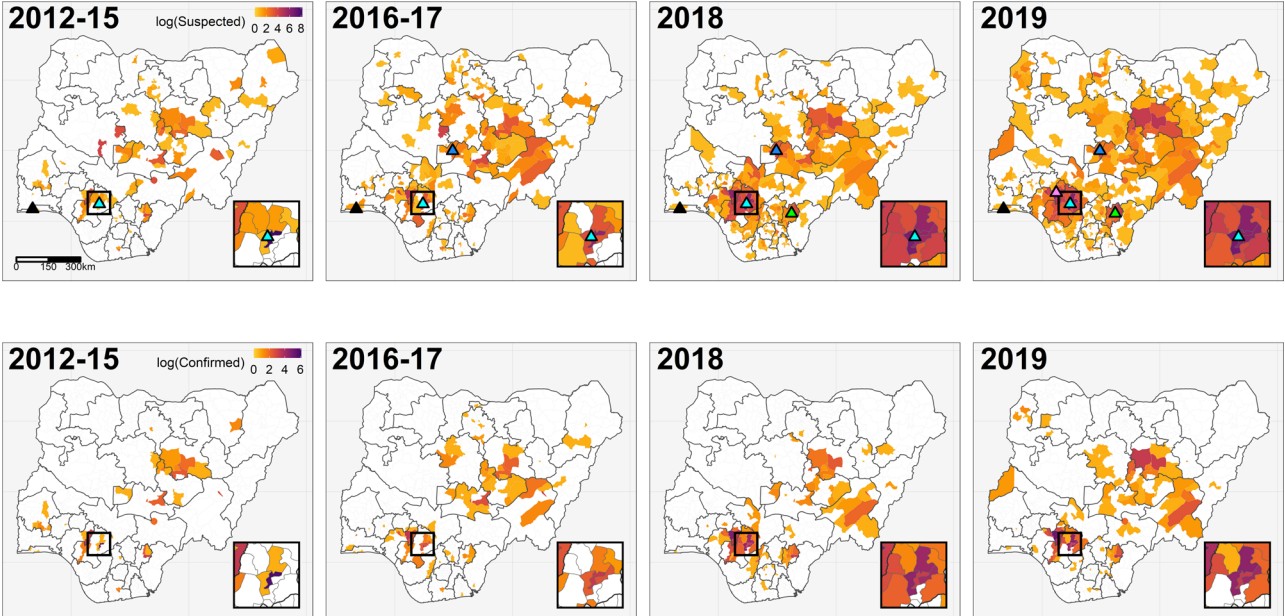

**Fig. 2 Spatiotemporal trends in Lassa fever surveillance, confirmed cases, and diagnostic laboratory capacity across Nigeria.** Maps show, on the natural log scale, the total reported Lassa fever cases (suspected and confirmed; top row) and laboratory-confirmed cases only (bottom row) in each local government authority during the specified year(s). Triangles in the top row show the locations of laboratories with Lassa fever diagnostic capacity. Irrua Specialist Teaching Hospital (Edo, established 2008; inset box, pale blue) and Lagos University Teaching Hospital (Lagos, southwest; black) were both operational since before 2012. Three further laboratories became operational during the study period: Abuja National Reference Laboratory in 2017 (FCT Abuja, north-central; dark blue), Federal Teaching Hospital Abakaliki in 2018 (Ebonyi, southeast; green), and Federal Medical Centre Owo in 2019 (Ondo, south, purple).

laboratories in 2017–19 (to a total of five; Fig. 2), the ongoing rollout of country-wide intensive training on LF surveillance, clinical case management and diagnosis (Supplementary Table 1)[22], and the deployment of a mobile phone-based reporting system to 18 states during 2017[23] (Methods). The result of these improvements is potentially reflected in the smoother case accumulation curves in 2018–19 than observed previously (Supplementary Fig. 1), as well as the notable, marked increase in the geographical extent of LF case reports over time. From 2012 to 2015 most reported cases originated from Esan Central in Edo state, the location of Nigeria's longest-established LF diagnostic laboratory and treatment centre at Irrua Specialist Teaching Hospital (ISTH)[19,20] (Fig. 2). The geographical extent of suspected and confirmed case reports rapidly expanded across Nigeria from 2016, with a contemporaneous decline in observed cases from Esan Central. This process can be seen clearly in LGAs surrounding Esan Central (Fig. 2, inset) and may reflect increasingly precise attribution of the true geographical origin of cases.

**Evaluating the geographical distribution and correlates of LF occurrence and incidence.** We developed spatiotemporal Bayesian models to evaluate the influence of climatic and socioeconomic factors on the geographical distribution of LF risk, using confirmed case data from 2016 to 2019 inclusive (i.e. the period following the rapid expansion of systematic surveillance; $n = 3096$; 774 LGAs over 4 years). We adopt a two-level, hurdle model-based approach, and separately model the annual probability of LF occurrence (using logistic regression, i.e. identifying the determinants of the LF endemic area; Fig. 3a) and incidence (using a zero-inflated Poisson likelihood, i.e. identifying the determinants of relative incidence within the endemic region; Fig. 3b). Country-wide surveillance has continued to expand since 2016, so we account for this ongoing expansion trend by fitting

annual, LGA-specific, spatially structured and unstructured random effects (Methods, Supplementary Fig. 2). For each response variable (occurrence and incidence) we conducted model selection by comparing candidate models including covariates to a spatiotemporal random effects only (baseline) model using Deviance Information Criterion (DIC). We considered linear and nonlinear effects for climate (several temperature and precipitation metrics) and linear effects for socioeconomic, landscape and reporting-based covariates (see Methods). Full selected models including covariates substantially improved overall model fit relative to baseline models for both occurrence ($\Delta DIC = -161.1$) and incidence models ($\Delta DIC = -195.2$) (Supplementary Table 2) and were robust to structured sensitivity tests (Methods, Supplementary Fig. 3).

Both full occurrence and incidence models included linear effects of agricultural land use, poverty prevalence and built-up land (Fig. 3c) and a nonlinear effect of mean annual precipitation (Fig. 3d, e). Additionally, the occurrence model included a negative effect of annual mean temperature, and the incidence model included a negative effect of travel time to the LF diagnostic laboratory (Fig. 3c). The results show that LF risk is strongly constrained by local climatic conditions, peaking in areas with medium-to-high annual precipitation levels (around 1500–2000 mm/year), and declining sharply in the more arid northeast, mirroring earlier work assessing the host environmental niche area (Redding et al. 2016). LF occurrence and incidence are also positively associated with increasing agricultural land use, which may synergistically affect both reservoir host population sizes and contact with people. Similarly there is a positive association with poverty and urbanisation, which likely jointly influence effective human–rodent contact, healthcare access and LF awareness (Fig. 3c and Supplementary Table 3). The strong positive association with built-up land appears counterintuitive given that LF has typically been considered a rural disease; this effect may be indexing other ecological (e.g.

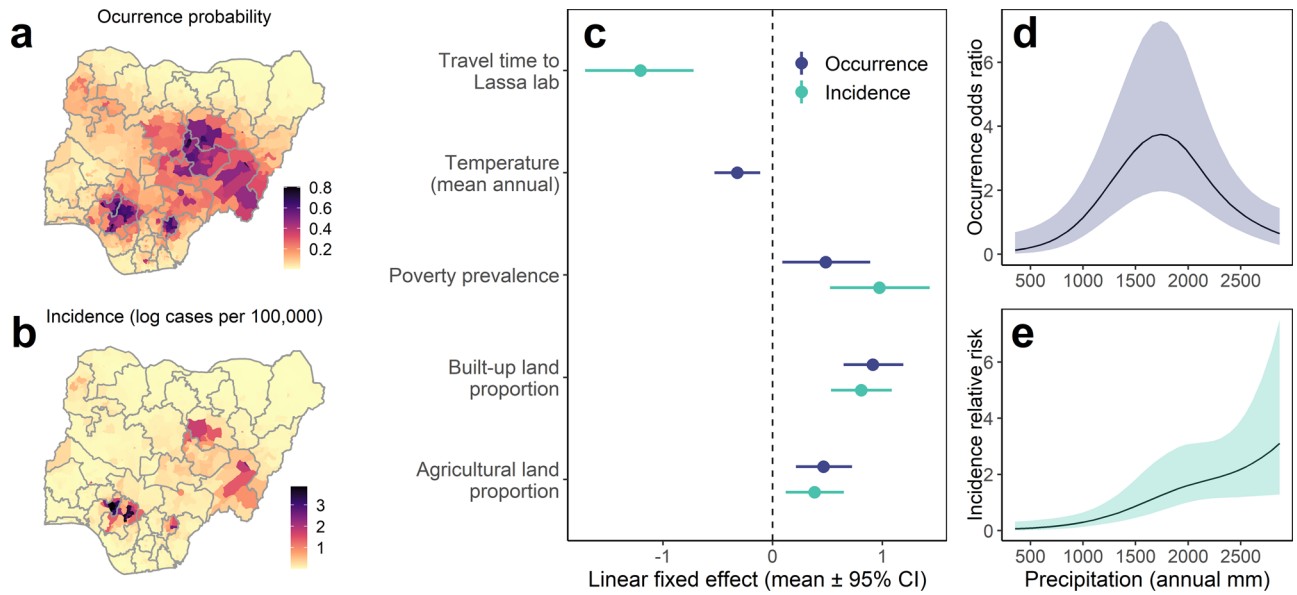

**Fig. 3 Spatial distribution and correlates of annual Lassa fever occurrence and incidence (2016 to 2019) at local government authority level across Nigeria.** Maps show fitted probability of LF occurrence (**a**) and incidence (**b**; cases per 100,000 persons, visualised on the natural log scale) for 774 LGAs in 2019. Points and error-bars (**c**) show socio-ecological linear fixed-effects parameter estimates (posterior mean and 95% credible interval) for best-fitting models of Lassa fever occurrence (dark blue; log odds scale) and incidence (pale green, log scale) ($n = 3096$ observations). Linear covariates were centred and scaled before fitting, so parameters measure the effect of 1 scaled unit change in the covariate (1 standard deviation) on either log odds of occurrence or log incidence. Curves show nonlinear effects of total annual precipitation on LF occurrence (odds ratio; **d**) and incidence (relative risk; **e**), specified and fitted as second-order random walks. Models included spatiotemporally structured random effects (LGA per year) to account for geographical heterogeneity and expansion of reporting effort (Methods) and were robust to cross-validation tests (Supplementary Fig. 3) and modelling at lower spatial resolution (Supplementary Fig. 4).

rodent synanthropy) or reporting-based processes (e.g. greater awareness and medical access in urbanised areas) that are not accounted for by other covariates. These results are consistent when modelling at lower spatial resolution (from 774 LGAs into 130 aggregated districts; Supplementary Fig. 4), although with weaker effects of urbanisation and poverty (suggesting these may act on detection or risk at more localised scales; Fig. 3b).

Overall, the limits of the endemic area of LF appear to be defined, therefore, by the interface of suitable environmental conditions for the reservoir host and socioeconomic conditions that facilitate human–reservoir contact, principally rainfed agricultural systems. Notably, LF incidence within endemic areas is higher in areas close to LF diagnostic laboratories (suggesting a substantial effect of detection bias) and is robustly associated with increasing poverty (Fig. 3c and Supplementary Fig. 3), which suggests that socioeconomic factors regulating human exposure are also an important driver of relative risk within environmentally suitable areas. Public programmes aimed at poverty alleviation, improving housing and sanitation infrastructure, and reducing human–rodent contact, may therefore have a positive impact in terms of reducing LF incidence.

Spatially projecting the fixed effects of the occurrence model (Supplementary Fig. 5) suggests that large contiguous areas of Nigeria are environmentally suitable for LF transmission and that underreporting may be highest in northern and eastern states. Some localities with high predicted suitability and nearby to existing endemic foci represent key areas to target increased surveillance (e.g. in Oyo, Osun and Ogun states). However, the socio-environmental covariates included in our models do not explain the highly discontinuous observed spatial distribution of LF (Fig. 3a, b) or the consistent, the very high incidence in Edo and Ondo states (Supplementary Figs. 2b, d and 5). One interpretation of our results is that the known geographical distribution of LF is predominantly shaped by surveillance effort

and that undetected cases are much more ubiquitous than currently recognised; this is plausible given the widespread nature of *M. natalensis*, and indeed is supported by the year-on-year expansion of the endemic area in Nigeria as surveillance continues to be rolled-out[1]. Additionally, since LASV prevalence in rodents can vary widely over small geographical scales (e.g. between neighbouring villages)[24], it is also possible that LF risk is highly discontinuous and localised. For example, fluctuations in rodent populations, pathogen dispersal, infection and immune dynamics, could lead to significant and challenging-to-predict variations in spillover risk over space and time. To identify underreported areas and target interventions, therefore, future surveys outside known endemic foci are urgently needed to understand unmeasured social or environmental factors influencing risk (e.g. high public and clinical awareness, agricultural practices[25] or LASV hyperendemicity in rodents[26,27]).

**Climatic predictors of seasonal LF peaks and the scope for forecasting.** Understanding and predicting how LF risk dynamics vary seasonally within the currently-identified endemic area is critical to inform disease diagnosis, prevention and control. We, therefore, analysed the spatiotemporal climatic predictors of weekly incidence (2012–2019) focusing on the six states that comprise the main endemic foci of LF in the south (Edo, Ondo and Ebonyi; 75% of all reported cases) and north (Bauchi, Plateau and Taraba; 12% of cases)[5,28]. We modelled weekly confirmed case incidence at the state-level as a Poisson process with state population as an offset ($n = 2820$; 470 weeks in six states; Methods). We first developed a baseline spatiotemporal model including spatially structured (state-level) and region-specific temporally-structured (year and week) random effects, and a fixed effect for travel time to Lassa diagnostic laboratory, to account for expansions of surveillance, intra-annual seasonality and baseline differences between states (Methods). To investigate

**Table 1 Retrospective and prospective predictive accuracy of climate-driven Lassa fever incidence model.**

| Model | RMSE (2012–2019) | RMSE (2016–2019) | RMSE (2020) | DIC (WS) |
|---|---|---|---|---|
| Baseline | 3.287 | 4.328 | 3.932 | 5030.5 |
| Climate-driven | 2.915 | 3.483 | 4.518 | 4793.9 |
| Proportion change in prediction error | −0.113 | −0.195 | 0.149 | |

The table shows the differences in the predictive performance of weekly LF cases by the baseline (random and reporting effects only) and best climate-driven model. We measured predictive performance as OOS RMSE, calculated for retrospective predictions across the entire study period (2012–2019) and following the widespread rollout of surveillance (2016–2019), and for prospective predictions for 2020. The climate-driven model also substantially improved within-sample model fit, measured using Deviance Information Criteria (DIC).

additional effects of environmental conditions on interannual LF outbreak dynamics and evaluate the scope for forecasting, we conducted out-of-sample (OOS) based model selection for linear and nonlinear effects of climate covariates: air temperature, vegetation greenness (Enhanced Vegetation Index; EVI), mean daily precipitation and Standardised Precipitation Index (SPI; a measure of relative drought or wetness in a 3-month window relative to historical trends at the same location). These were averaged across a 60-day period starting at 0, 1, 2, 3 and 4-month lags prior to reporting week, to account for delayed effects of climate and any delays between infection and reporting (Methods, Supplementary Fig. 6). We considered candidate models for all lagged combinations of all four covariates, and identified the model that minimised OOS predictive error (measured as root mean square error, RMSE) on sequential 6-month holdout windows across the study period (Table 1).

The best combination of climatic predictors included nonlinear effects of SPI (120–180 days lag), precipitation (60–120 days lag) and EVI (0–60 days lag), and substantially reduced OOS error relative to a random and reporting effects-only baseline model (11.3% reduction in RMSE over 2012–2019, and 19.5% reduction over 2016–2019; Table 1). Posterior predictive simulation for sequential 6-month holdout windows showed that the climate-driven model had a good ability to reproduce historical case trends (Fig. 4a, b and Supplementary Figs. 7, 8; 93.5% of observations falling within the 95% predictive interval). Separately examining the marginal effects of year, season and climate covariates over time suggests that combinations of interannual changes in reporting, natural seasonality and climatic factors, explain both LF periodicity (distance between peaks) and trends (relative height of peaks over time) in both regions (Supplementary Fig. 7). Climatic variability is associated with interannual differences in predicted timing and amplitude of LF seasonality. The climate-driven relative risk does appear to have been unusually high in the south during the large case surge in 2018, although climate conditions during recent high-incidence years overall tend to fall within a similar range to previous years (Supplementary Fig. 6a). This suggests that the unprecedented surges in 2018–2019 probably resulted mainly from a sharp change in surveillance and/or other unmeasured factors (Supplementary Fig. 7b).

The climate-driven model results suggest that seasonal LF risk in endemic areas is linked to the distribution of rainfall in the preceding months (non-extreme conditions at 120–180 days lag, and low precipitation at 60–120 days lag) and with declines in vegetation in the preceding 0–60 days (Fig. 4c–e). Together with the geographical models of annual incidence (Fig. 3c–e), these results point to a substantial effect of climate in explaining LF occurrence and incidence patterns across Nigeria. The effect of lagged rainfall and vegetation dynamics on seasonal risk strongly suggests an important role of reservoir host population ecology. Indeed, past studies have shown that inter- and intra-annual precipitation characteristics—including the distribution of rainfall throughout the rainy season—are predictive of subsequent *M. natalensis* population surges and crop damage in East Africa[29].

Temporal variation in rodent and human LASV infections may be driven by seasonal and interannual rodent population dynamics (putatively linked to climate-driven cycles in resource availability and land use[30]) or human agricultural and food storage practices[4], all of which are important targets for future research. For example, declines in vegetation during the weeks preceding transmission could lead to synchronous food-seeking behaviour in rodents (as natural food availability reduces) and human behaviour changes relating to harvest and crop processing. The retrospective OOS predictive accuracy of the seasonal models (Fig. 4 and Supplementary Fig. 8) suggests that, provided reporting effort is adequately accounted for, lagged climate variables might feasibly assist in advance forecasting of LF peaks a month in advance within known endemic areas[31]. To examine this further, we used the baseline and climate-driven models fitted to 2012–2019 data to make prospective predictions of weekly cases beyond the study period (to 31 December 2020), fixing all effects except climatic predictors at 2019 levels (i.e. assuming that reporting effort and other interannual differences stay the same). Both models substantially underpredict the true number of cases observed in 2020 (841 climate-driven and 798 baseline, compared to 1027 reported), potentially because neither effectively captures ongoing improvements in surveillance sensitivity or other potential unobserved events (e.g. hospital-acquired infections). The climate-driven model predicts a higher number of cases overall, likely because the model slightly better anticipates the longer duration of the Lassa season in 2020 than the baseline model (Fig. 4 and Supplementary Fig. 9). However, the climate-driven model shows a worse prospective predictive performance on weekly observations than the baseline (15% increase in OOS RMSE; Table 1), which may be a consequence of the relatively short time series used to learn climate associations ($n = 8$ years, and only 3 years following the rollout of surveillance).

## Discussion

Our results show that LF is a climate and land-use sensitive disease of poverty. Much of the environmental mediation of case numbers is likely driven by ecological dependencies of the principal host *M. natalensis*, resulting in a strong association between LF cases and rainfall. Peak LF occurrence appears to be in areas experiencing around 1500 mm of rain annually, though incidence increases with rainfall, though with very wide uncertainty under very high rainfall conditions. While the role of host ecological suitability could be important, it may also be human behaviours that vary in response to climatic conditions, such as crop planting or farming techniques, that could be the principal or additional underlying cause of case variation. The trend of increasing incidence with higher rainfall, but the decreasing probability of disease occurrence, could be reflecting the correlation between rainfall and host habitat suitability in Nigeria. Areas with very high rainfall in the south are principally rainforest, which is unsuitable habitat for *M. natalensis*[4]. One possible implication for disease risk is that landscape conversion towards derived savannah or agriculture in these areas could drive increases in risk, as

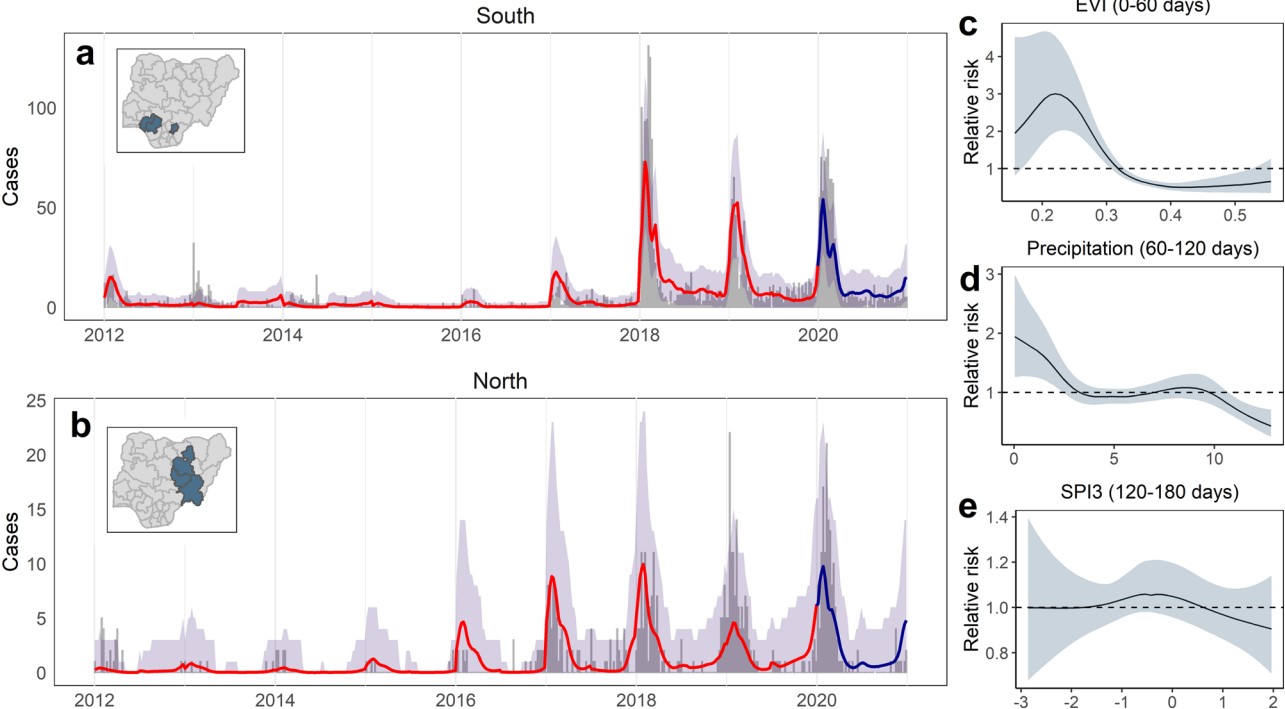

**Fig. 4 Modelled temporal dynamics and drivers of confirmed Lassa fever cases in the south and north Nigeria.** Case time series show observed and out-of-sample (OOS) predicted weekly case counts from a climate-driven model (n = 2820), summed across all states in the southern (**a**; Edo and Ondo states) and northern (**b**; Bauchi, Plateau and Taraba states) endemic areas to visualise regional differences. Time series graphs (**a**, **b**) show observed counts from 2012 to 2019 (grey bars), OOS posterior median predicted cases (red line) and OOS 95% (grey shading) posterior predictive intervals (both calculated from 2500 samples drawn from the joint posterior). OOS predictions were made while holding out sequential 6-month windows across the full time series at state-level (Supplementary Fig. 8). The dark blue line and shaded area in 2020 shows prospective predicted cases (median and 95% predictive interval) compared to observed cases from this period, which were not included in model fitting (Methods). Inset maps show states included in the models. Panels show nonlinear fitted effects of Enhanced Vegetation Index (EVI) (**c**), mean daily precipitation (**d**), and 3-month Standardised Precipitation Index (SPI3; **e**) on relative risk, showing posterior mean and 95% credible interval. The marginal contributions of yearly, seasonal and climatic effects are visualised separately in Supplementary Fig. 7.

environmental conditions become more conducive for *M. natalensis* populations.

While the broad limits of the endemic area of LF occurrence and seasonal nature of observed LF cases may be driven by the ecological tendencies of the host or climate-responsive agricultural practices, the spatially discontinuous nature of incidence could be driven by several factors, for example, land-use patterns creating a mosaic of suitable and unsuitable host habitat, or heterogeneous human socioeconomic factors. Although our models identify an overall role of poverty, this is a higher level, compound metric that is likely to influence several more specific risk factors, such as the prevalence of rodent hunting, availability of food storage options, or the probability of having poor-quality housing that allows high levels of rodent ingress. The environmental covariates we examine in our models are, however, still limited in their ability to account for the fine-scale, LGA-level heterogeneity in case incidence; in future, incorporating more detailed socioeconomic and anthropological data may provide further insights. Additionally, it could be that reporting effort, which our models account for using random and distance-based effects, is driving the variation in observed cases at these finer spatial scales, and that more detailed data on reporting procedures would provide clearer insights into these spatial patterns.

Indicative of this, the highest reporting state, Edo, is not predicted to be a very high incidence state using socio-environmental factors alone, with high cases mostly explained by the effect of distance to the laboratory. From the comparison of the yearly maps (Fig. 2), there is an apparent over-reporting of cases in Esan

Central (the location of Irrua Specialist Teaching Hospital) in the early years of the dataset, while in later years the geographical locations of cases in Edo become more spatially disaggregated as surveillance and reporting systems are improved. This role of reporting effort in explaining case numbers is generally suggestive of missing cases in areas typically considered to be low incidence. The high case-fatality rates in lower reporting areas[1] could be a further indication that less severe cases are going undetected. Efforts to better delimit the endemic area and total burden of LF would strongly benefit from investigation of areas with medium-to-high predicted suitability but currently lower observed case incidence, to understand if these are truly low incidence locations.

During the time period of data collection, West Africa experienced two particularly disruptive epidemics: the Ebola epidemic in 2013–15, and the ongoing global Covid-19 pandemic. Observed cases during the period of the Ebola epidemic are overestimated by our predictive models (Fig. 4) suggesting that our models are not accurately capturing some aspects of the climate variation, or potentially that case reporting was lower due to behavioural changes (e.g. hospital avoidance) or repurposed diagnostic capacity. Our ability to characterise the predictors of interannual LF variation is, as such, limited by the relatively short length and substantial interannual variation of our case time series, and by the need to account for an ongoing trend of increasing surveillance.

The improvement we show in retrospective predictive performance provided by a climate-driven model suggests that such an approach could in the future provide the basis for developing a

forecast system in LF-endemic areas (similar to systems increasingly used for dengue[32]). However, the uncertain prospective performance shows that robust evaluation of climate-based forecasting approaches will be limited by the length and quality of the case time series used to develop the models. Improving these models year-on-year with new surveillance data, and including more precise information on agricultural practices, spatiotemporal variation in landscape characteristics and modelled rodent suitability, should assist in further identifying the drivers of large case surges and improving prospective prediction accuracy. Overall our findings have implications for ongoing disease management and targeting of LASV surveillance in rodents and humans toward environmentally suitable areas where LF is apparently absent, prioritising hotspot areas for future vaccination rollouts, and highlights the critical role of improvements in systematic human case surveillance across West Africa[20,21] in helping to explain rising numbers of cases. With LF surveillance continuing to improve in Nigeria, these data and analyses provide a foundation for the future development of user-focused LF risk mapping and forecasting systems, to aid public health responses in this region.

## Methods

We analyse weekly reported counts of suspected and confirmed human cases and deaths attributed to LF (as defined in Supplementary Table 1), between 1 January 2012 and 30 December 2019, from across the entire of Nigeria. The weekly counts were reported from 774 LGAs in 36 Federal states and the Federal Capital Territory, under Integrated Disease Surveillance and Response (IDSR) protocols, and collated by the NCDC. All suspected cases, confirmed cases and deaths from notifiable infectious diseases (including viral haemorrhagic fevers; VHFs) are reported weekly to the LGA Disease Surveillance and Notification Officer (DSNO) and State Epidemiologist (SE). IDSR routine data on priority diseases are collected from inpatient and outpatient registers in health facilities, and forwarded to each LGA's DSNO using SMS or paper form. Subsequently, individual LGA DSNOs collate and forward the data to their respective SE, also by SMS and paper form, for weekly and monthly reporting respectively to NCDC. From mid-2017 onwards, data entry in 18 states has been conducted using a mobile phone-based electronic reporting system called mSERS, with the data entered using a customised Excel spreadsheet that is used to manually key into NCDC-compatible spreadsheets. Data from this surveillance regime (WERs) were collated by epidemiologists at NCDC throughout the period 2012 to March 2018 (Supplementary Fig. 1).

Throughout the study period, within-country LF surveillance and response has been strengthened under NCDC coordination[2,20,33]. LGAs are now required to notify immediately any suspected case to the state-level, which in turn reports to NCDC within 24 h, and also sends a cumulative weekly report of all reported cases. A dedicated, multi-sectoral NCDC LF TWG was set up in 2016 with the responsibility of coordinating all LF preparedness and response activities across states. Further capacity building occurred in 2017 to 2019, with the opening of three additional LF diagnostic laboratories in Abuja (Federal Capital Territory), Abakaliki (Ebonyi state) and Owo (Ondo state) (to a total of five; Fig. 2) and the rollout of intensive country-wide training on surveillance, clinical case management and diagnosis. We note that, due to the rapid expansion in a test capacity, the definition of a suspected case in our data has subtly changed over the surveillance period: from 2012 to 2016, suspected cases include probable cases that were not lab-tested, whereas from 2017 to 2019, all suspected cases were tested and confirmed to be negative.

In addition to the WERs data, since 2017 LF case reporting data has also been collated by the LF TWG and used to inform the weekly NCDC LF Situation Reports (SitRep data; https://ncdc.gov.ng/diseases/sitreps). This regime includes post hoc follow-ups to ensure more accurate case counts, so our analyses use WER-derived case data from 2012 to 2016, and SitRep-derived case data from 2017 to 2019 (see Fig. 1 for full time series). A visual comparison of the data from each separate time series, including the overlap period (2017 to March 2018) is provided in Supplementary Fig. 1, and all statistical models considered random intercepts for the different surveillance regimes. Where other studies of recent Nigeria LF incidence have been more spatially and temporally restricted[34,35], the extended monitoring period and fine spatial granularity of these data provide the opportunity for a detailed empirical perspective on the local drivers of LF at a country-wide scale and their relationship to changes in reporting effort.

### Recent trends in LF surveillance in Nigeria.

We visualised temporal and seasonal trends in suspected and confirmed LF cases within and between years, for both surveillance datasets. Weekly case counts were aggregated to country-level and visualised as both annual case accumulation curves, and aggregated weekly case totals (Fig. 1 and Supplementary Fig. 1). We also mapped annual counts of

suspected and confirmed cases across Nigeria at the LGA-level to examine spatial changes in reporting over the surveillance period (Fig. 2). State and LGA shapefiles used for modelling and mapping were obtained from Humanitarian Data Exchange under a CC-BY-IGO license (https://data.humdata.org/dataset/nga-administrative-boundaries).

Analyses of aggregated district data are sensitive to differences in scale and shape of aggregation (the modifiable areal unit problem; MAUP[36]), and LGA geographical areas in Nigeria are highly skewed and vary over >3 orders of magnitude (median 713 km², mean 1175 km², range 4–11,255 km²). We therefore also aggregated all LGAs across Nigeria into 130 composite districts with a more even distribution of geographical areas, using distance-based hierarchical clustering on LGA centroids (implemented using hclust in R), with the constraint that each new cluster must contain only LGAs from within the same state (to preserve potentially important state-level differences in surveillance regime). Weekly and annual suspected and confirmed LF case totals were then calculated for each aggregated district. We used these spatially aggregated districts to test for the effects of scale on spatial drivers of LF occurrence and incidence.

### Statistical analysis.

We analysed the full case time series (Fig. 1) to characterise the spatiotemporal incidence and drivers of LF in Nigeria, while controlling for year-on-year increases and expansions of surveillance effort. We firstly modelled annual LF occurrence and incidence at a country-wide scale, to identify the spatial, climatic and socio-ecological correlates of disease risk across Nigeria. Secondly, we modelled seasonal and temporal trends in weekly LF incidence within hyperendemic areas in the north and south of Nigeria, to identify the seasonal climatic conditions associated with LF risk dynamics and evaluate the scope for forecasting. All data processing and modelling was conducted in R v.3.4.1 with the packages R-INLA v.20.03.17[37], raster v.3.4.13[38] and velox v0.2.0[39]. Statistical modelling was conducted using hierarchical regression in a Bayesian inference framework (integrated nested Laplace approximation (INLA)), which provides fast, stable and accurate posterior approximation for complex, spatially and temporally-structured regression models[37,40], and has been shown to outperform alternative methods for modelling environmental phenomena with evidence of spatially biased reporting[41].

*Processing climatic and socio-ecological covariates.* We collated geospatial data on socio-ecological and climatic factors that are hypothesised to influence either *M. natalensis* distribution and population ecology (rainfall, temperature and vegetation patterns), frequency and mode of human–rodent contact (poverty and improved housing prevalence), both of the above (agricultural and urban land cover) or likelihood of LF reporting (travel time to nearest laboratory with LF diagnostic capacity and travel time to nearest hospital). For each LGA we extracted the mean value for each covariate across the LGA polygon. The full suite of covariates tested across all analyses, data sources and associated hypotheses are described in Supplementary Table 5.

We collated climate data spanning the full monitoring period and up until the date of analysis (July 2011 to January 2021). We obtained daily precipitation rasters for Africa[42] from the Climate Hazards Infrared Precipitation with Stations (CHIRPS) project; this dataset is based on combining sparse weather station data with satellite observations and interpolation techniques, and is designed to support hydrologic forecasts in areas with poor weather station coverage (such as tropical West Africa)[42]. A recent study ground-truthing against weather station data showed that CHIRPS provides greater overall accuracy than other gridded precipitation products in Nigeria[43]. Air temperature daily minimum and maximum rasters were obtained from NOAA and were also averaged to calculate daily mean temperature. EVI, a measure of vegetation quality, was obtained from processing 16-day composite layers from NASA (National Aeronautics and Space Administration) (excluding all grid cells with unreliable observations due to cloud cover and linearly interpolating between observations to give daily values; Supplementary Table 5).

We derived several spatial bioclimatic variables to capture conditions across the full monitoring period (Jan 2012 to Dec 2019): mean precipitation of the driest annual month, mean precipitation of the wettest annual month, precipitation seasonality (coefficient of variation), annual mean air temperature, air temperature seasonality, annual mean EVI and EVI seasonality. We also calculated monthly total precipitation, 3-month SPI[44], average daily mean ($T_{mean}$), minimum ($T_{min}$) and maximum ($T_{max}$) temperature and EVI variables at sequential time lags prior to reporting week for seasonal modelling (described below in Temporal drivers). SPI is a standardised measure of drought or wetness conditions relative to the historical average conditions for a given period of the year. SPI was calculated within a rolling 3-month window across the full 40-year historical CHIRPS rainfall time series (1981–2020) using the R package SPEI v.1.7[44].

We accessed annual human population rasters at 100 m resolution from WorldPop. We accessed the proportion of the population living in poverty in 2010 (<$1.25 threshold) from WorldPop, to proxy for ability to access risk prophylaxis schemes (e.g. food storage boxes) and for potential susceptibility to disease as a consequence of lower nutrition and co-infection. We accessed modelled proportion of the population living in improved housing in 2015[45], to proxy for the potential for homes to be infested with rodents. We accessed data on agricultural and urban land cover (population-weighted proportion of LGA area) for 2015 from processing ESA-CCI rasters.

Finally, we used a global travel friction surface and a least-cost path algorithm[46] to calculate LGA-level mean travel time to the nearest LF diagnostic laboratory (as a proxy for likelihood of sample testing for LASV) and nearest hospital[47] (to proxy for the probability of patients accessing healthcare when unwell). Such distance-based metrics are coarse approximations of complex processes and are subject to limitations. For example, differences in access to transport infrastructure and political unrest will have different effects on reporting in different areas of Nigeria, regardless of proximity to medical facilities, and clinical suspicion for LF will also be influenced by staff training and sensitisation. Furthermore, diagnostic centres are often established in areas where the disease is already recognised to occur (e.g. in Owo in 2019; Fig. 2), so the direction of causality is unclear. The ongoing rollout of electronic reporting systems should in the coming years provide extra information on the role of reporting in determining LF case patterns.

*Evaluating the geographical distribution and correlates of LF occurrence and incidence.* We modelled annual LF occurrence and incidence at a country-wide scale to determine the spatial, socio-ecological correlates of disease across Nigeria. We used annual confirmed case counts per-LGA across the last 4 years of surveillance (2016 to 2019) as a measure of LF incidence, since these years followed the establishment of updated systematic surveillance protocols and the associated geographical expansion of suspected case reports (Fig. 2), and so are likely to more fully represent the true underlying distribution of LF across Nigeria. In total, 161 LGAs reported confirmed LF cases from 2016 to 2019, with the majority of cases reported from a much smaller subset (75% from 18 LGAs), and 613 LGAs reported no confirmed LF cases (total = 774 LGAs; median 0 cases, mean 2.21, range 0–321). This overdispersed and zero-inflated distribution presents a challenge for fitting to incidence counts, so we instead adopt a two-stage, hurdle model-based approach, and separately model LF occurrence in all LGAs using logistic regression, and incidence using a zero-inflated Poisson likelihood (which models zero observations as a mixture of true and false negatives). Previous iterations of these analyses had used a zero-truncated negative binomial model for incidence; instead using a zero-inflated model provided the benefit of retaining all the data, as well as improving goodness of fit. This both ensures that fitted models adhere to distributional assumptions, and also enables a clearer separation of the contributions of different socio-ecological factors to disease occurrence (i.e. the presence of LF) and to total case numbers in endemic areas.

We model the annual occurrence of LF ($n = 774$ LGAs over 4 years) where $Y_{i,t}$ is the binary presence (1) or absence (0) of LF in LGA $i$ during year $t$, and $p_{i,t}$ denotes the probability of LF occurrence, such that:

$$Y_{i,t} \sim \mathrm{Bern}\left(p_{i,t}\right) \tag{1}$$

We model annual LF case counts ($C_{i,t}$) as a zero-inflated Poisson process, where $z$ is a parameter describing the probability of observing a zero count and $\mu_{i,t}$ is the expected number of cases in LGA $i$ during year $t$, such that:

$$P\left(C_{i,t} = c\right) = z \cdot 1_{[c=0]} + (1 - z)\frac{\mu_{i,t}{}^{c}e^{-\mu_{i,t}}}{c!} \tag{2}$$

Both $p_{i,t}$ and $\mu_{i,t}$ are separately modelled as functions of socio-ecological covariates and random effects based on the general linear predictor:

$$\mathrm{logit}\left(p_{i,t}\right) = \alpha + \sum_j \beta_j X_{j,i} + \sum_k \delta_{k,i} + u_{i,t} + v_{i,t} \tag{3}$$

$$\log\left(\mu_{i,t}\right) = \alpha + P_{i,t} + \sum_j \beta_j X_{j,i} + \sum_k \delta_{k,i} + u_{i,t} + v_{i,t} \tag{4}$$

where, for each model, $\alpha$ is the intercept; $X$ is a matrix of climatic and socio-ecological covariates with linear coefficients given by $\beta$; $\delta_{k,i}$ are nonlinear effects for climatic predictors (specified as second-order random walks); and spatiotemporal reporting trends at LGA level are accounted for using annual spatially structured (conditional autoregressive; $v_{i,t}$) and unstructured i.i.d. (independent and identically distributed) ($u_{i,t}$) random effects jointly specified as a Besag–York–Mollie model. The incidence model additionally includes log human population in each year ($P_{i,t}$) as an offset. We set penalised complexity priors for all random effects hyperparameters, and uninformative Gaussian priors for fixed effects.

For both models we considered linear coefficients ($\beta$) for the following covariates: mean precipitation of the driest month, mean precipitation of the wettest month, precipitation seasonality, annual mean air temperature, temperature seasonality, annual mean EVI, EVI seasonality, proportion agricultural land cover, proportion urban land cover, the proportion of the population living in poverty (<$1.25 per day), the proportion of the population living in improved housing and two distance-based covariates to account for reporting effort: mean travel time to the laboratory with LF diagnostic capacity and mean travel time to the nearest hospital. We also considered nonlinear (random walk) terms for temperature and rainfall covariates because past studies of *M. natalensis* distribution suggest that these responses may be nonlinear[12,13]. Prior to modelling we removed covariates that were highly collinear with one or more other others (Pearson correlation coefficient >0.8). Continuous covariates not log-transformed were scaled (to mean 0, s.d. 1) prior to fitting linear fixed effects.

We conducted model inference and selection in R-INLA, and evaluated model fit for both occurrence and incidence models using DIC[48,49]. We conducted model selection on fixed effects by comparing to a random effects-only spatiotemporal baseline model. For temperature and precipitation variables, we first decided whether to consider linear or nonlinear effects by sequentially fitting each covariate as either linear or nonlinear, and selecting the variable that minimised DIC. We then conducted full selection on all covariates by removing each in turn from a full model (including all covariates), and excluding any that did not improve fit by a threshold of at least six DIC units. The best-fitting models with socio-ecological covariates explained substantially more of the variation in the data relative to baseline models (occurrence $\Delta$DIC = −161.1; incidence $\Delta$DIC = −195.2; Supplementary Table 2). All posterior parameter distributions and residuals were examined for adherence to distributional assumptions. We evaluated the contribution of socio-ecological effects to predicted LF occurrence and incidence by examining the difference in LGA-level random effects between baseline and full models[50] (Supplementary Fig. 2) and by spatially projecting fixed effects across Nigeria (Supplementary Fig. 5).

We evaluated the sensitivity of spatial model results to geographically-structured cross-validation, in turn fitting separate models holding out all LGAs from each of 12 states that have either high (Bauchi, Ebonyi, Edo, Nasarawa, Ondo, Plateau and Taraba) or low (Kogi, Delta, Kano, Enugu and Imo) documented incidence. Fixed and nonlinear effects direction and magnitude were robust in all holdout models, indicating that results were not overly driven by data from any one locality (Supplementary Fig. 3). We also tested for sensitivity to aggregation scale (i.e. MAUP) by refitting the final occurrence and incidence models to the data aggregated into 130 approximately equal-sized districts (as described above). Confirmed LF case totals were calculated for each district, socio-ecological covariates were extracted and models were fitted as described above (Supplementary Fig. 4 and Supplementary Table 3).

*Climatic predictors of seasonal LF peaks and the scope for forecasting.* A growing body of data from clinical records[6,19,21], ethnographic and social science research[10,51] and rodent population and serological monitoring[9,52] suggests that LF risk may be climate-sensitive. Temporal trends in human and rodent infection are hypothesised to be associated with seasonal cycles in rodent population ecology, human land use and food storage practices[4]. We, therefore, developed spatio-temporal models to quantify the lagged climatic and environmental conditions that predict LF incidence (weekly case counts) across the full duration of surveillance (2012 to 2019). Low and/or variable surveillance effort outside known endemic areas could confound inference of temporal environmental drivers, so here we focus our analyses on states with case reporting records that span the entire monitoring period. These occur in two foci in the south (Edo, Ondo and Ebonyi states) and north regions of Nigeria (Bauchi, Plateau and Taraba states), which in total account for 87% of the total confirmed cases since 2012 (Fig. 3). These regions (north and south) are distinct in terms of agro-ecologies and climate (Supplementary Fig. 6)[53], so models included spatially structured and region-specific temporally-structured random effects to account for these differences.

We fitted models to state-level LF time series from these six states (Supplementary Table 4). Although our source data are at fine (LGA) resolution, modelling seasonal climate associations at coarser state-level scale better harmonises the resolution of disease data with climatic data, and reduces potential noise associated with uncertain attribution of the true LGA of origin for cases in the early part of the time series (especially in Southern states; see Fig. 2). We model weekly case counts $Z_{i,t}$ ($n = 6$ states over 8 years, so total of 2820 observations) as a Poisson process:

$$Z_{i,t} \sim \mathrm{Pois}\left(\mu_{i,t}\right) \tag{5}$$

where $\mu_{i,t}$ is the expected number of cases for state $i$ during week $t$, modelled as a log-link function of a linear combination of spatially and temporally-structured random effects and climate covariates:

$$\log\left(\mu_{i,t}\right) = \alpha + P_{i,t} + \gamma_{r(i),t} + \rho_{r(i),t} + u_{i,t} + v_{i,t} + \sum_j \beta_j X_{j,i} + \sum_k \delta_{k,i} \tag{6}$$

Here, $\alpha$ is the intercept, $P_{i,t}$ is log human population included as an offset (thereby modelling incidence), and several random effects are included to account for space and time: $\gamma_{r(i),t}$ is a region-specific temporally-structured effect of the year (first-order random walk fitted separately for north and south, to account for ongoing changes in reporting effort and other interannual differences), $\rho_{r(i),t}$ is a region-specific seasonal effect of the epidemiological week to account for seasonality (second-order random walk to capture dependency between weeks, fitted separately for north and south), and $u_{i,t}$ and $v_{i,t}$ are state-level spatially structured and unstructured (i.i.d.) random effects jointly specified as a BYM model, as above. Additionally, $X$ is a matrix of covariates (including travel time to diagnostic laboratory) with linear coefficients given by $\beta$, and $\delta_{k,i}$ are nonlinear effects of climatic predictor variables (specified as second-order random walks). We set penalised complexity priors for all hyperparameters and uninformative Gaussian priors for fixed effects. A Poisson model with seasonal random effects was sufficient to account for seasonal overdispersion in the data without using a negative binomial likelihood.

We conducted model selection to identify the model that minimised OOS predictive error on sequential holdout windows across the study time series. This involved fitting 16 sub-models for each candidate model, each holding out all observations in a 6-month window at a time (January–June or July–December of each year), and extracting OOS predicted case counts for the holdout window. A model predictive error was calculated as RMSE of the difference between observed and OOS predicted case counts across the whole time series (2012–2019). We first conducted this procedure for a baseline model containing only random (state, season and year) and reporting effects (travel time to laboratory), which was used as a benchmark to compare against climate-driven model performance (Table 1).

To identify the combination of climate variables that minimised predictive error, we then conducted the same procedure for candidate models containing all combinations of four climate predictors at five different time lags: mean daily precipitation, SPI, EVI and mean daily air temperature, calculated across a 60-day window at time lags beginning from 0 days to 120 days prior to reporting week (i.e. 0–60, 30–90, 60–120, 90–150 and 120–180 days; spanning 0 to 6 months before reporting). We considered lagged climate variables to account for, firstly, delayed effects of seasonal environmental cycles on *M. natalensis* population ecology, behaviour and LASV prevalence that are hypothesised to influence the force of infection to humans[9], and secondly, delays between LASV infection event, disease incubation period (which can be up to 10 days[4]) and patient presentation at a medical facility. We included both precipitation and SPI as these reflect different, biologically relevant hydrometeorological phenomena: precipitation is a raw measure of rainfall, whereas SPI measures drought or wetness relative to historical trends at the same location and period of the year (and thus reflects deviations from average expected rainfall)[44]. We did not include the temporally-invariant covariates included in the spatial models, since the smaller number of states provides low comparative power to detect any spatial effects on incidence.

We then examined the calibration of the best climate-driven model through posterior predictive simulation, again using 6-month sequential holdout windows. Each sub-model was fitted, 2500 parameter samples were drawn from the approximated joint posterior distribution, and these were used to (1) calculate OOS posterior mean and intervals and (2) simulate the OOS Poisson predictive distribution (i.e. the range of plausible expected case counts given the model). We calculated the proportion of observed case counts falling within 67 and 95% OOS predictive intervals, overall and over time (Supplementary Fig. 8).

Finally, to evaluate the scope for model-based prospective forecasting, we used the baseline and climate-driven model to make prospective predictions of posterior mean case counts and predictive intervals for the whole of 2020 (using climate data up to December 2020) from the model fitted to the full time series. We compared these predictions to 2020 preliminary state-wide confirmed case counts compiled for the NCDC Situation Reports, holding yearly random effects ($\gamma_{r,t}$) at the same level as 2019 (i.e. predictions for 2020 assume the same level of effort; Fig. 4 and Supplementary Fig. 9). Because these are preliminary data they are unsuitable for model fitting but provide a useful future OOS test for prospective forecasting ability.

**Reporting Summary**. Further information on research design is available in the Nature Research Reporting Summary linked to this article.

## Data availability

All data used for these analyses are provided at the accompanying repository https://doi.org/10.6084/m9.figshare.9777656. Social and environmental covariate datasets are openly available online, and links are provided in Supplementary Table 5.

## Code availability

All code used for these analyses are provided at the accompanying repository https://doi.org/10.6084/m9.figshare.9777656.

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

## Acknowledgements

The authors thank all the epidemiologists and clinical staff at Nigeria state and LGA levels who collected and submitted the original case reports. This research was supported by an MRC UKRI/Rutherford Fellowship (MR/R02491X/1) and Sir Henry Dale Research Fellowship (funded by the Wellcome Trust and the Royal Society) (220179/Z/20/Z) (both D.W.R.), a Graduate Research Scholarship (R.G.) and Global Engagement Fund grant (D.W.R. and R.G.) both from University College London and the QMEE CDT, funded by NERC grant number NE/P012345/1 (L.A.A.). C.A.D. acknowledges joint Centre funding from the UK Medical Research Council and Department for International Development (DFID). C.A.D. acknowledges joint Centre funding from the UK Medical Research Council and Department for International Development (DFID). C.A.D. is funded by the Department of Health and Social Care using UK Aid funding on a grant managed by the UK National Institute for Health Research (NIHR) (Vaccine Efficacy Evaluation for Priority Emerging Diseases: PR-OD-1017-20007 and HPRU in Emerging and Zoonotic Infections: NIHR200907). The views expressed in this publication are those of the author(s) and not necessarily those of the NHS, the National Institute for Health Research or the Department of Health and Social Care. I.A. acknowledges funding from the UK NIHR (NF-SI-0616–10037), EDCTP PANDORA Consortium and the UK MRC. K.E.J. acknowledges the Dynamic Drivers of Disease in Africa Consortium, NERC project no. NE-J001570-1, which was funded with support from the Ecosystem Services for Poverty Alleviation Programme (ESPA). The ESPA programme was funded by DFID, the Economic and Social Research Council (ESRC) and NERC.

## Author contributions

C.C.D.-N., E.A.I., Y.R.U., O.H.S., A.O.M., I.A. and C.I. collected the data; D.W.R., R.G., L.A., C.A.D., I.A., K.E.J. and C.I. designed the study; R.G. led and conducted the analyses with D.W.R. and L.A.A. and R.G., D.W.R., C.A.D. and K.E.J. wrote the manuscript. All authors contributed to writing and editing the full manuscript.

## Competing interests

The authors declare no competing interests.
