## [Peer Review File · Nature Communications]

REVIEWER COMMENTS

Reviewer #1 (Remarks to the Author):

The authors have gathered an impressive data set and done an interesting analysis.

The statistical methods are difficult to evaluate. The authors' approach allows a huge number of possible models, so the only possible validation is via out-of-sample comparisons. The authors recognize this, and report both "holdout" and "prospective" comparisons. They do not explain their standard for evaluating these comparisons.

I have two concerns with the holdout comparisons. First, they are not really OOS, as far as I can tell. The authors choose their model from a broad combinatoric array, and then apparently keep the same model structure for all of the holdouts. In other words, the data held out has a strong effect on the model, via variable choice. It seems to me plausible that this might not be a problem, but equally plausible that it might. The authors need to discuss this problem and provide evidence that it's not a concern based on either null simulations or citation to a methodological study. Otherwise, it looks like they're making up a very complicated new validation technique.

Second, the held out numbers fall inside the 95% CI substantially more than 95% of the time. The authors deal with this concern at some length in a response to review, but not in the MS.

The prospective comparisons are not supported by summary statistics, but only by a "visual" inspection. The summary statistics must be given in the main text.

The regional models are hard to evaluate. Most of the meat is in ST3, which (presumably) has cryptic units. The authors should do an appropriate rescaling and make a figure for the fixed parameters, at least. It is difficult to tell from the text (§ at L212) whether the vegetation effects have the same sign (in fact they don't). The s. at L220 is particularly problematic: the reader who hasn't dived into the supplementary table will almost certainly think that the authors found *_similar_* effects across two different regions.

The authors need to say more about the different combinations and approaches that they tried. Much of their modeling is based on two hot spots, which are modeled separately. If this is the only combination that they tried, they should say so explicitly. If they tried other approaches, or even carefully considered them while looking at data patterns, they should say so (and of course also feel free to explain why they settled on the choices that they made).

The authors talk several times about a NB model for case counts, and several times about a Poisson model for case counts. I was not able to figure out if this is for slightly different applications, or if it is just a typo. If it's not a typo, the authors' need to justify this difference in choices. In either case, they need to clarify and/or correct.

The authors do not adequately explain the role of "total reported cases" to control for effort in their incidence models – it doesn't seem to be addressed in the Methods at all?

L150 "predicted solely" is misleading. It is not predicted perfectly, nor do we accept the null hypothesis that the other predictors have no value (there was just none seen clearly in this particular study). The authors should say that it was the only variable selected, or showing a clear effect, or crossing a significance threshold.

L156 The fact that the two spatial scales provide similar information seems more like a sanity check than like "important" information.

L166 "mainly explained" seems too strong (sort of conflates the model with reality). I feel like that whole clause could be skipped (just say "poorly predicted")

Two reasonable alternatives are discussed around L172. The authors should also explicitly mention the possibility that both effects are at work

Jonathan Dushoff

Reviewer comments:

Reviewer #1 (Remarks to the Author) - Jonathan Dushoff.

The authors have gathered an impressive data set and done an interesting analysis.

The statistical methods are difficult to evaluate. The authors' approach allows a huge number of possible models, so the only possible validation is via out-of-sample comparisons. The authors recognize this, and report both "holdout" and "prospective" comparisons. They do not explain their standard for evaluating these comparisons.

Thank you for your useful perspectives and for highlighting these issues. We have conducted a substantial reanalysis to improve the clarity and interpretability of the statistics, which we hope will have addressed your concerns. To summarise, we have re-designed all modelling throughout the manuscript so that each model that includes climatic/socio-ecological effects is explicitly compared to the performance of a 'baseline' random-effects only model. This approach is increasingly widespread in spatiotemporal climate-disease modelling (e.g. Lowe et al 2016, 2021) and provides a clear benchmark against which to compare either the explanatory ability or predictive performance of environmentally-driven models.

For the explanatory models of annual LF occurrence and incidence (Figure 3), the baseline random effects-only model is a full spatiotemporal model, with annual spatially structured and unstructured effects at LGA-level that explicitly capture the changing geography of surveillance effort over time (shown in Extended Data Figure 2). Model selection for climatic and socioecological covariates was conducted based on minimising DIC, and the differences in fit between baseline and socio-ecological models are now clearly stated in the main text (lines 179-186). The updated approach of comparing to a baseline model, explicitly assessing annual shifts in LF surveillance geography, and changing the incidence model to a zero-inflated likelihood to better handle uncertainty in zero observations (discussed below), has provided several benefits. Firstly, we identify much clearer, nonlinear relationships to climate (Figure 3), as well as linear socio-ecological effects, which remain robust to geographical sensitivity tests (Extended Data Figure 3). Secondly, comparing between baseline and full models allows clearer visualisation of the geographical areas in which the socio-ecological and climate covariates contribute most to explaining LF patterns (by showing where the random effects shrink towards 0 in the full model; Extended Data Figure 2b).

For the predictive models of temporal LF dynamics in endemic states (Figure 4), the baseline model is a random and reporting effects-only model that includes spatial, yearly, seasonal and lab distance-based effects. The model comparisons and selection for best combination of climatic effects are now conducted entirely out-of-sample (OOS; as discussed below), with the best climate-driven model chosen based on minimising OOS root mean square error (RMSE) on 6-month holdout windows for the full study period. We then evaluate whether climate improves predictive ability by directly comparing the OOS predictive performance of the best climate-driven model to that of the baseline model (using RMSE) with metrics shown in full in an additional Table 1 in the main text. This shows that, for retrospective holdout-based prediction, including climate information (nonlinear effects of precipitation, Standardised Precipitation Index and EVI) substantially reduces predictive error relative to the baseline model (Table 1, lines 495-503).

We also take the same approach to the prospective predictions for 2020 (now including the full year of Lassa case data), which interestingly shows that the climate-driven model does not clearly improve prospective prediction performance over the baseline model (main text 304-313). This comparative approach with baseline model as benchmark has therefore

provided a fuller demonstration of the potential strengths but also current limitations (especially around the short case time series) of forecasting approaches for Lassa fever, which we address more in an expanded Discussion to suit the format of *Nature Communications*.

With these latest findings in mind, and the paper's aim to both explain spatial patterns and evaluate potential scope for future forecasting, we have also changed the title of the manuscript to better reflect our revised approach and findings: 'Spatiotemporal analysis reveals geographical drivers and predictable climate-linked dynamics of Lassa fever in Nigeria'.

I have two concerns with the holdout comparisons. First, they are not really OOS, as far as I can tell. The authors choose their model from a broad combinatoric array, and then apparently keep the same model structure for all of the holdouts. In other words, the data held out has a strong effect on the model, via variable choice. It seems to me plausible that this might not be a problem, but equally plausible that it might. The authors need to discuss this problem and provide evidence that it's not a concern based on either null simulations or citation to a methodological study. Otherwise, it looks like they're making up a very complicated new validation technique.

We agree with this concern that the hybrid approach, we took for pragmatic reasons, could be improved. We have therefore updated the model selection to be fully out-of-sample: we fit 16 sub-models and estimate OOS RMSE for candidate models containing all possible combinations of 4 predictors at multiple time lags (temperature, EVI, daily precipitation, and an additional drought indicator, SPI, that reflects deviations from a location's expected historical precipitation trend, so its biological relevance is more directly comparable across different geographical locations). Because the updated spatial modelling identified clear, nonlinear effects of precipitation, we considered both linear and nonlinear effects in the model selection. Addressing the concern about modelling separate regions, we combined both regions into a single model of state-level incidence that included random effects for regional differences (discussed further below). The selected best model contained nonlinear effects of 3 predictors: SPI at 120-180 day lag; Precipitation at 60-120 day lag; and EVI at 0-60 day lag, and substantially improved OOS performance relative to a baseline (Table 1).

Second, the held out numbers fall inside the 95% CI substantially more than 95% of the time. The authors deal with this concern at some length in a response to review, but not in the MS.

By combining the two regions and conducting the full model selection out-of-sample, this issue has lessened somewhat in the updated models: now, overall, 93.5% of observations fall in the 95% interval from the climate model, although predictions overall remain somewhat under-dispersed relative to observed cases. We have been clearer about the ramifications of this in a revised Discussion (lines 263-279, 347-383).

The prospective comparisons are not supported by summary statistics, but only by a "visual" inspection. The summary statistics must be given in the main text.

We have provided a full Table 1 in the main text giving summary statistics of both retrospective holdout and prospective differences in predictive error (OOS RMSE) for climate-driven and baseline models. We hope this provides clear numerical evidence for improvements provided by climate information (as well as being clearer about the ambiguity in the prospective predictions).

The regional models are hard to evaluate. Most of the meat is in ST3, which (presumably) has cryptic units. The authors should do an appropriate rescaling and make a figure for the

fixed parameters, at least. It is difficult to tell from the text (¶ at L212) whether the vegetation effects have the same sign (in fact they don't). The s. at L220 is particularly problematic: the reader who hasn't dived into the supplementary table will almost certainly think that the authors found _similar_ effects across two different regions.

The authors need to say more about the different combinations and approaches that they tried. Much of their modeling is based on two hot spots, which are modeled separately. If this is the only combination that they tried, they should say so explicitly. If they tried other approaches, or even carefully considered them while looking at data patterns, they should say so (and of course also feel free to explain why they settled on the choices that they made).

Thank you for this perspective and we agree with your concerns about the ambiguity of the findings for the different regions. The decision to conduct the modelling into two regions was originally made because the climate dynamics and agroecology are somewhat different. However, since we do not expect the effects of climate and vegetation dynamics on fundamental ecological processes (i.e. rodent population ecology) to differ wholly between these two areas, it is probably more appropriate to pool all the data in a single model to learn these associations.

We have taken this approach, combining the two regions and including region-specific seasonal and year effects to account for differences in climate and reporting trends. We have also moved to a slightly coarser spatial scale and now conduct the temporal modelling at the state-level (6 high-incidence states), because this is directly aligned with the preliminary 2020 state-level case counts, ensuring an appropriate exact comparison for the prospective prediction tests (Extended Data Figure 9).

This updated approach has shown clear and biologically plausible nonlinear effects of vegetation and precipitation indicators (suggesting, for example, that higher risk is associated with low vegetation greenness, but that this appears to decline below a threshold level that potentially reflects resource limitations for rodents in the far north; Figure 4c). It is feasible that these nonlinearities may have been partially responsible for the ambiguous results in the linear fixed effects for the previous subregion-level models. These results are now presented both in Supplementary Tables (SI Table 2), and in Figure 4 on the relative risk scale, so their units are easier to interpret.

The authors talk several times about a NB model for case counts, and several times about a Poisson model for case counts. I was not able to figure out if this is for slightly different applications, or if it is just a typo. If it's not a typo, the authors' need to justify this difference in choices. In either case, they need to clarify and/or correct.

This was a typo from an earlier version of the analyses; all incidence models are based on a Poisson model and we have corrected this throughout the manuscript.

The authors do not adequately explain the role of "total reported cases" to control for effort in their incidence models – it doesn't seem to be addressed in the Methods at all?

This was included originally to attempt to adjust for the changing geography of overall surveillance effort in the incidence models, although we agree this was not clearly explained. In the updated manuscript, we have instead addressed the problem of surveillance expansion more simply and tractably by fitting full spatiotemporal random effects (i.e. year specific effects at LGA-level) to explicitly model geographical trends in LF surveillance (Extended Data Figure 2), and by changing the incidence likelihood to a zero-inflated Poisson to better handle uncertainty in zero counts (i.e. accounting for the likely presence of false negatives in under-surveyed areas) (Lines 693-699). This approach is more in line with

convention and also makes it easier to interpret the contribution of random and reporting-based effects to the model (see Extended Data Figure 2).

L150 "predicted solely" is misleading. It is not predicted perfectly, nor do we accept the null hypothesis that the other predictors have no value (there was just none seen clearly in this particular study). The authors should say that it was the only variable selected, or showing a clear effect, or crossing a significance threshold.

This is a good point, and we do not use this term anymore. The specific issue this refers to (a single predictor being selected for the incidence model) has also changed with updates to the spatial models: the updated incidence model now identifies several other predictors as robustly identified with incidence as well as occurrence. Notably, more explicitly modelling the geographical expansion of surveillance using random effects alongside the new OOS model selection has resulted in simpler and more intuitive models being selected.

L156 The fact that the two spatial scales provide similar information seems more like a sanity check than like "important" information.

We agree this makes sense; we have therefore moved the coarser-scale model into the Supplementary Material and discussed it briefly in the main text.

L166 "mainly explained" seems too strong (sort of conflates the model with reality). I feel like that whole clause could be skipped (just say "poorly predicted")

We have updated this sentence to address your point and to reflect the results of the updated models (lines 223-226).

Two reasonable alternatives are discussed around L172. The authors should also explicitly mention the possibility that both effects are at work

This is a good point - it is indeed likely that both effects are at work in the Nigeria context. We have changed and edited much of this section to more clearly explain the context and highlight that these are non-exclusive explanations (lines 188-217).

REVIEWERS' COMMENTS

Reviewer #1 (Remarks to the Author):

This is a good work and I certainly like the changes.

I think that the authors need to be even more careful about their statistical conclusions, however. Comparing many models and using holdouts is a good method for developing predictive methods (although it didn't work out well here, the authors discuss this part well). It is not easy to use this method to test hypotheses like: do these results show a clear effect of climatic drivers on incidence. The fact that they gambled on prediction but did not do well in 2020 means that a lot of questions are still up in the air.

I don't think that this is a major problem, as long as the authors can tone down their conclusions still further. The authors outline a potentially important data set and ask good questions. Given all of the issues with the data, it's not too surprising that they don't see much clearly, and there is reason to hope that the data will keep improving and that the authors have laid a useful foundation.

The authors need to be open about statistical models; how did they converge on these methods and what other things did they try (e.g., negative binomial fits)?

Line references are to the markup MS (the only one I seem to have access to).

L59 The authors are talking here about "observed" incidence.

L68 "may be" potential seems better supported

L156 implications seem a little strong. Is it really "apparent" that these phenomena reflect diagnosis and reporting rather than spread? Cf. L164.

The clause on L198 and the s on L200 step on each other a bit; they could be edited more smoothly

Around L230, the authors could talk more about dynamic effects (rodent population fluctuations, local introductions and extinction of disease, accumulation of rodent immunity); the fundamentally dynamical nature of infectious disease tends to make it less predictable than non-infectious disease.

"Step change" is unclear. "Sharp"?

L360 (and probably elsewhere): clearer to say "observed" cases

Jonathan Dushoff

We thank the reviewer and editor for their recommendations and revised the manuscript to address all the raised points. Please find in red our responses below:

Reviewer comments:

Reviewer #1 (Remarks to the Author) - Jonathan Dushoff.

This is a good work and I certainly like the changes.

I think that the authors need to be even more careful about their statistical conclusions, however. Comparing many models and using holdouts is a good method for developing predictive methods (although it didn't work out well here, the authors discuss this part well). It is not easy to use this method to test hypotheses like: do these results show a clear effect of climatic drivers on incidence. The fact that they gambled on prediction but did not do well in 2020 means that a lot of questions are still up in the air.

I don't think that this is a major problem, as long as the authors can tone down their conclusions still further. The authors outline a potentially important data set and ask good questions. Given all of the issues with the data, it's not too surprising that they don't see much clearly, and there is reason to hope that the data will keep improving and that the authors have laid a useful foundation.

We thank the reviewer for their time and constructive comments that have substantially improved this manuscript. We have toned down the final conclusions to meet the suggestions of the reviewer (lines 348-351) and addressed further specific points regarding this below. We have only included line numbers where they are not included in the reviewer's statement.

The authors need to be open about statistical models; how did they converge on these methods and what other things did they try (e.g., negative binomial fits)?

We have added more detail to methods to describe this process (lines 706-711 & 830-832).

L59 The authors are talking here about "observed" incidence.

We have added in the word "observed" as suggested.

L68 "may be" potential seems better supported
We have changed this as requested.

L156 implications seem a little strong. Is it really "apparent" that these phenomena reflect diagnosis and reporting rather than spread? Cf. L164.
We have changed this to "potentially reflected in".

The clause on L198 and the s on L200 step on each other a bit; they could be edited more smoothly

This is a good point; we have altered this to be two short sentences instead

Around L230, the authors could talk more about dynamic effects (rodent population fluctuations, local introductions and extinction of disease, accumulation of rodent immunity); the fundamentally dynamical nature of infectious disease tends to make it less predictable than non-infectious disease.

We have added in this extra detail as request (lines 235-237)

"Step change" is unclear. "Sharp"?

We have change this to "sharp" (Line 284).

L360 (and probably elsewhere): clearer to say "observed" cases

added in the word "observed" as suggested and edited the rest of text to add in "observed" where is not clear we are talking about reported cases or not.